# MicroRNA-21 Expression as a Prognostic Biomarker in Oral Cancer: Systematic Review and Meta-Analysis

**DOI:** 10.3390/ijerph19063396

**Published:** 2022-03-14

**Authors:** Mario Dioguardi, Francesca Spirito, Diego Sovereto, Mario Alovisi, Giuseppe Troiano, Riccardo Aiuto, Daniele Garcovich, Vito Crincoli, Luigi Laino, Angela Pia Cazzolla, Giorgia Apollonia Caloro, Michele Di Cosola, Lorenzo Lo Muzio

**Affiliations:** 1Department of Clinical and Experimental Medicine, University of Foggia, Via Rovelli 50, 71122 Foggia, Italy; spirito.francesca97@gmail.com (F.S.); diego_sovereto.546709@unifg.it (D.S.); giuseppe.troiano@unifg.it (G.T.); elicio@inwind.it (A.P.C.); dott.dicosola@gmail.com (M.D.C.); lorenzo.lomuzio@unifg.it (L.L.M.); 2Department of Surgical Sciences, Dental School, University of Turin, 10127 Turin, Italy; mario.alovisi@unito.it; 3Department of Biomedical, Surgical and Dental Science, University of Milan, 20122 Milan, Italy; riccardo.aiuto@unimi.it; 4Department of Dentistry, Universidad Europea de Valencia, Paseo de la Alameda 7, 46010 Valencia, Spain; daniele.garcovich@universidadeuropea.es; 5Department of Basic Medical Sciences, Neurosciences and Sensory Organs, Division of Complex Operating Unit of Dentistry, “Aldo Moro” University of Bari, Piazza G. Cesare 11, 70124 Bari, Italy; vito.crincoli@uniba.it; 6Multidisciplinary Department of Medical-Surgical and Odontostomatological Specialties, University of Campania “Luigi Vanvitelli”, 80121 Naples, Italy; luigi.laino@unicampania.it; 7Unità Operativa Nefrologia e Dialisi, Presidio Ospedaliero Scorrano, ASL (Azienda Sanitaria Locale) Lecce, Via Giuseppina Delli Ponti, 73020 Scorrano, Italy; giorgiacaloro1983@hotmail.it

**Keywords:** oral cancer, miR-21, microRNA, OSCC, HNSCC, non-coding RNA

## Abstract

Oral carcinoma represents one of the main carcinomas of the head and neck region, with a 5-year survival rate of less than 50%. Smoking and tobacco use are recognized risk factors. Prognostic survival biomarkers can be a valid tool for assessing a patient’s life expectancy and directing therapy towards specific targets. Among the biomarkers, the alteration of miR-21 expression in tumor tissues is increasingly reported as a valid prognostic biomarker of survival for oral cancer. The purpose of this meta-analysis was, therefore, to investigate and summarize the results in the literature concerning the potential prognostic expression of tissue miR-21 in patients with OSCC. Methods: The systematic review was conducted following the PRISMA guidelines using electronic databases, such as PubMed, Scopus, and the Cochrane Central Register of Controlled Trials, with the use of combinations of keywords, such as miR-21 AND oral cancer, microRNA AND oral cancer, and miR-21. The meta-analysis was performed using the RevMan 5.41 software. Results: At the end of the article-selection process, 10 studies were included in the meta-analysis, and the result for the main outcome was a pooled HR per overall survival (OS) of 1.29 (1.16–1.44) between high and low expression of miR-21. Conclusions: The data in the literature and the results emerging from the systematic review indicate that miR-21 can provide a prognostic indication in oral cancer.

## 1. Introduction

Oral squamous cell carcinoma (OSCC) accounts for 90% of oral cancers and is the cancer with the highest incidence in the head and neck region. OSCC is histologically characterized by the presence of squamous cells in the multi-layered epithelium that lines the oral cavity [1].

More than 350,000 new cases of OSCC are diagnosed every year worldwide, but the poor prognosis of oral cancer has not improved significantly in recent decades [2], with a tendency to worsen even for the young population [3]. The 5-year survival rate of patients with oral cancer is less than 50% [4], and the therapies applied, including surgery, radiotherapy, and chemotherapy, lead to a significant reduction in the quality of life [5,6].

The most important risk factors for the development of oral cancer are tobacco and alcohol use, and it is estimated that approximately 80% of oral cancers occur due to these factors [7,8]. Other risk factors have been identified concomitantly with other pathologies, such as the presence of oral precancerous diseases (leukoplakia, erythroplakia, and lichen planus) and infectious agents (HPV, HCV, and EBV) [9,10,11,12]. Moreover, traumatic events may contribute to the pathogenesis of oral cancer, along with nutritional factors, genetic factors, ultraviolet radiation, and immunosuppression [1].

The main genetic alterations affecting carcinogenesis involve changes in tumor suppressors (APC, p53 BRCA2, PTCH, NF1, VHL, Rb BCL2, SWI/SNF, p16, CD95, ST5, YPEL3, ST7, and ST14) and oncogenes (Ras, jun, fas, erbA, abl, raf, gsp, sis, erbB, and fms) in addition the onset and progression of oral cancer involve changes of DNA methylation, histone modifications, and non-coding alterations of RNA, including microRNAs (miRNA) [13,14,15,16,17,18,19,20,21].

MicroRNAs (miRNAs) are a family of small endogenous non-coding RNAs that regulate a series of biological processes by influencing gene expression, the dysregulation of which allows the tumor cell to develop tumor capacities, including increased proliferative signaling with the avoidance of suppressive mechanisms, resistance to apoptosis, invasive and metastatic capacities, and the induction of angiogenesis [22].

A previous meta-analysis by our research group identified how the alteration of miRNA expression in tissues could be used as a predictor of worse prognosis in patients with oral cancer [23]. In particular, the downregulated miRNAs were miR-204, miR-101, miR-32, miR-20a, miR-16, miR-17, and miR-125b, while those upregulated with prognostic capability were miR-21, miR-455-5p, miR-155-5p, miR-372, miR-373, miR-29b, miR-1246, miR-196a, and miR-181. Data in a recent literature review also confirmed miR-21, miR-27a (-3p), miR-31, miR-93, miR-134, miR-146, miR-155, miR-196a, miR-196b, miR-211, miR-218, miR-222, miR-372, and miR-373 as upregulated and let-7a, let-7b, let-7c, let-7d, let-7e, let-7f, let-7g, let-7i, miR-26a, miR-99a-5p, miR-137, miR-139-5p, miR-143-3p, miR-184, and miR-375 as downregulated in oral cancer [24,25,26,27].

In particular, miR-21 appears to be promising as a potential diagnostic biomarker for head and neck tumors and for OSCC [26,28].

The human miR-21 gene has 3433 nucleotides located in an intergenic zone on chromosome 17q23.2 [29]. The miR-21 gene has two transcription sites: T1 (the minor transcription site) and T2 (the primary transcription site). At the nucleus level, the RNA pol II enzyme transcribes primary miR-21 (~3.5 kb), which is transported to the cytoplasmic site, where polymerase III releases the 22-nucleotide miR-21, which is incorporated and degraded by the miRNA–RISC complex. In tumor cells, there may be changes in the bioregulation of miR-21 attributable to alterations in the mechanisms of expression, transcription, transport, binding with the RISC, and degradation [30,31]

The prognostic power of the tissue expression of miR-21 for oral cancer was investigated in recent studies by Supic et al., who identified a hazard ratio between low expression and high expression of miR-21 for OS (overall survival) of 2.002 (0.904–4.434) [32], which is not in agreement with Jakob et al., who identified an HR of 0.16, and also not concordant with the results from a cohort of patients of the TCGA (HR: 3.99; data extrapolated by Jakob et al.) [33]. The latest systematic review with a meta-analysis was conducted by Xie and Wu [34], specifically on the prognostic power of miR-21 for OSCCs, identifying an aggregate HR of 1.7 (1.20–2.44) for OS. New studies on the prognostic value of miR-21 in oral carcinoma highlight that it is essential to have a hazard ratio result that is as precise and up to date as possible.

The purpose of this meta-analysis was, therefore, to investigate and summarize the results in the literature regarding the prognostic potential of the altered expression of tissue miR-21 in patients with OSCC.

## 2. Materials and Methods

### 2.1. Protocol and Registration

The systematic review was conducted following the PRISMA guidelines [35]; the protocol, the inclusion criteria, the search strategy, and the search outcomes were specified and recorded before the search and screening of the articles on PROSPERO with the registration number CRD42022301429.

### 2.2. Eligibility Criteria

All prospective and retrospective studies and RCTs that evaluated differences in the tissue expression of miR-21 in oral carcinoma in correlation with prognostic survival indices were considered, with the following points identified: participants (patients with oral cancer), intervention (altered expression of miR-21 in OSCC), control (patients with oral cancer who have low expression of miR-21 in OSCC), and outcome (difference in the prognosis of survival among patients with low and high miR-21 expression in OSCC). The PICO question, therefore, was as follows: is there a difference in the prognostic survival rates between patients with oral cancer with high miR-21 expression and those with low expression?

The inclusion criteria were as follows: studies that indicated and reported data on prognostic survival indices (hazard ratios, Kaplan–Meier curves, and Cox regression) between low and high miR-21 expression in oral carcinoma tissues in patients with the disease.

The exclusion criteria were as follows: studies published in a language other than English, those that did not present data on survival and the expression of miR-21, and those at high risk of bias.

### 2.3. Sources of Information, Research and Selection

The studies were identified through literature searches in electronic databases by two researchers (M.D. and D.S.). Limits relating to the language of publication were applied, and articles in a language other than English were excluded. The literature search was conducted on the PubMed, Scopus, and Cochrane databases. The last bibliographic search was conducted on 10 December 2021. In addition, a search of the gray literature and on Google Scholar was also conducted, and the bibliographic sources of previous systematic reviews on the subject were investigated.

We used the following terms to search the databases: miR-21 AND oral cancer, microRNA AND oral cancer, and miR-21. Duplicates were removed using EndNote and manually. The identified articles were independently evaluated and scrutinized by 2 reviewers (M.D. and D.S.), with title and abstract analysis for potentially eligible studies and text analysis for inclusion in the systematic review. Furthermore, the k agreement between the two reviewers was assessed, and a third reviewer resolved any situations of disagreement.

### 2.4. Data Collection Process and Data Characteristics

The data to be extracted were previously established by the 2 reviewers responsible for screening the studies and were reported in 2 tables independently to be subsequently compared to minimize the risk of errors. The data that were extracted from the articles concerned the type of study, the year of publication, the first author, the country that conducted the study, the number of patients, the type of OSCC, the type of miR investigated, the cut-off between low and high expression, the hazard ratio (HR), and the presence of a Kaplan–Meier curve. Specifically, all the HR data for high expression vs. low expression of miR-21 in tumor tissues (OSCC) concerning the overall survival (OS), disease-free survival (DFS), recurrence-free survival (RFS), and cancer-specific survival (CSS) were searched and extracted if the data were in the form of Kaplan–Meier curves.

### 2.5. Risk of Bias in Individual Studies, Summary Measures, Summary of Results, Risk of Bias between Studies, and Additional Measures

The risk of bias in the individual studies was assessed by one author (M.D.), with a second author tasked with verifying the correct assessment (D.S.). Parameters derived from the Reporting Recommendations for Tumor Marker Prognostic Studies (REMARK) were used for the assessment [36], and studies with a high risk of bias were excluded from the meta-analysis. The results, represented by forest plots and inconsistency indices such as the Higgins index *I*^2^, were evaluated.

The risk of bias between the studies was assessed graphically through the analysis of the overlaps of the confidence intervals, through the *I*^2^ inconsistency index (an *I*^2^ value greater than 75% was considered high, and a random-effects analysis was applied in specific cases), and through a funnel plot. If the meta-analysis presented high heterogeneity indices, a sensitivity analysis was carried out, excluding only the studies that presented a low overlap of the confidence intervals or that emerged graphically from the funnel plot.

For the meta-analysis, and particularly for the calculation of the pooled HR, the software Reviewer Manager 5.4 (Cochrane Collaboration, Copenhagen, Denmark) was used. We used the GRADE pro-Guideline Development Tool online software (GRADE pro-GDT, Evidence Prime, Hamilton, ON, Canada) to assess the quality of the evidence.

## 3. Results

### 3.1. Selection of Studies

The search in the Scopus, PubMed, and Cochrane Central Register of Controlled Trials databases provided 777 bibliographic citations. Following the removal of overlaps, 260 registrations were obtained, of which 140 were excluded because it was found that they did not meet the eligibility criteria upon reading the abstracts, and 120 articles were potentially admissible, but only eight met the inclusion criteria and were included in the meta-analysis. Furthermore, the analysis of the gray literature (Google Scholar and OpenGrey) and of the previous systematic reviews allowed the identification of two further studies for inclusion in the meta-analysis (Figure 1).

### 3.2. Data Characteristics

The articles included in the meta-analysis are as follows: Jung et al. (2012) [37], Kawakita et al. [38], Hedbäck et al. [39], Yu et al. [40], Supic et al. [32], Jakob et al. [33], Li et al. (2013) [41], Zheng et al. [42], Li et al. (2009) [43], and Ganci et al. [44].

The characteristics of the extracted data are described in the Materials and Methods section for the survival data expressed in the form of a Kaplan–Meier curve and HR and extracted according to the Tierney method [45]. All the extracted data are reported in Table 1.

All 10 included studies are retrospective clinical studies whose results were published between 2009 and 2019 and include a follow-up period of 5 to 15 years. The total number of oral cancer patients recruited is 708. The studies by Li et al. (2009) [43], Zheng et al. (2016) [42], Supic et al. [32], and Kawakita et al. [38] specifically considered OTSCC, while the remaining six studies considered OSCC more generally. The studies by Junget al. [37], Supic et al. [32], Jakob et al. [33], Zheng et al. [42], and Li et al. (2009) [43] considered OS as a prognostic parameter; Hedbäck et al. [39] and Yu et al. [40] used DFS; and RFS was used as a prognostic index by Jakob et al. [33] and Ganci et al. [44], while CSS is only indicated by Kawakita et al. [38].

The main outcomes researched and extracted from the studies were the prognostic indices of survival—specifically, the HR between high and low expression. Furthermore, in order to minimize the heterogeneity between the studies in the meta-analysis, the main outcome was divided according to the function of the different prognostic indices:Primary outcome: HR for OS between high and low miR-21 expression.Secondary outcome: HR for DFS between high and low miR-21 expression.Tertiary outcome: HR for CSS between high and low miR-21 expression.Quaternary outcome: HR for RFS between high and low miR-21 expression.

### 3.3. Risk of Bias in Studies

The risk of bias was assessed through parameters derived from the Reporting Recommendations for Tumor Marker Prognostic Studies (REMARK). Based on the REMARK guidelines, a score from 0 to 3 was considered for each factor (Table 2).

### 3.4. Primary Outcome and Secondary Outcome

The meta-analysis results for the first outcome of the HR for OS between high and low expression, represented by the forest plot in Figure 2, show a pooled HR of 1.29 (1.16–1.4); the heterogeneity is Chi^2^ = 12.41; df = 5 (*p* < 0.03); *I*^2^ = 60; test for the overall effect: Z = 4.72 (*p* < 0.00001).

The results of the meta-analysis for the second outcome, which included only two studies, from Hedbäck et al. [39] and Yu et al. [40], show an increased HR of 2 (1.35–2.95) for DFS; heterogeneity *I*^2^ = 0% is absent, Chi^2^ = 0.52; df = 1 (*p* = 0.45); test for overall effect (*p* = 0.0005) (Figure 3).

### 3.5. Tertiary and Quaternary Outcomes

For the third outcome (the HR for CSS between high and low miR-21 expression), only one study was included (Kawakita et al. [38]), and the data were extrapolated by the author from the Kaplan–Meier curve according to the method of Tierney. The HR between high and low miR-21 expression was found to be 1.19 (0.72–1.97) for CSS (Figure 4).

The fourth outcome took into consideration the RFS as a prognostic index, and the resulting pooled HR was 1.41 (0.48–4.15) with high *I*^2^ heterogeneity of 86%; in fact, it is notable that the confidence intervals of the two studies barely overlap. Conflicting data were reported for the RFS prognostic index (Figure 5).

### 3.6. Risk of Bias across Study

The risk of bias in the studies is low for the first and second outcomes (*I*^2^ values of 60 and 0%, respectively), and the funnel plot confirms the low heterogeneity for the first outcome, as no sources of heterogeneity were identified (Figure 6); for the third outcome, it was not possible to evaluate it, as only one study was included, while for the fourth outcome (the HR prognostic index RFS), there could be a bias in the study by Jakob et al. [33], determined by the low number of patients included by the authors.

We also used GRADE pro-GDT to assess the quality of the primary outcome (Table 3). The results suggest that the quality of the evidence is moderate for the primary outcome.

## 4. Discussion

Almost all of the included studies with different degrees of significance support the prognostic value of the different levels of miR-21 expression.

Jung et al. reported significant data from a study conducted on 17 patients, with lower survival rates in patients with high expression of miR-21: OS: HR, 5.31 (C.I.: 1.39–20.38). The limit of this study in the authors’ opinion, unfortunately, lies in the low number of carcinomas investigated, especially compared to those in other studies, which, however, partially confirm the results [37].

The study on CSS in squamous cell tumors of the tongue conducted by Kawakita et al. [38] is interesting, with data reporting that miR-21 tended to be particularly highly expressed in the invasive front of cancerous tissues, and the overexpression of miR-21 was detected in 60 out of 79 cases. In this case, the extracted HR value was calculated starting from the survival curve using the Tierney method, with the data extracted as follows: an HR of 1.19 (0.71–1.9), differing little from the data extracted from the previous review of Xie and Wu, which reported an HR of 1.16 for CSS [34].

Hedbäck et al. was among the first to link the localization of miR-21 in relation to the presence of myofibroblasts in OSCC and reported that the DFS had an HR of 2.7 between high and low expression [39].

In a study conducted on a cohort of 147 patients with OSCC, Yu et al. (2017) reported DFS in agreement with the data of Hedbäck et al., with an HR of 1.86, indicating how miR-21 may be an independent factor associated with disease-free survival in OSCC [40].

Data from Jakob et al. regarding RFS (HR, 0.18) contrast with the data present in the literature (Ganci et al.); RFS: HR, 4.20 [44]; motivation, as explained by the authors, should be sought in the small number of samples analyzed [33].

Upon extrapolating the data from the survival curve in the study by Zheng et al., a hazard ratio of 1.22 between high and low expression for the OS was obtained. The study revealed that there was also a correlation between miR-21 expression and chemoresistance [42], while Li et al. (2009) revealed how miR-21 plays a role in the development of malignancy, making it an unfavorable prognostic factor for OTSCC.

All these studies indicate that there is a prognostic value for miR-21 in oral cancer, high expression of which in the tissues affected by the neoplasia is characterized by a worsening of survival.

The present systematic review performed for OS reports an aggregate HR of 1.29. This value is partly in line with the results of a previous meta-analysis conducted by Xie and Wu [34], which reported a pooled HR of 1.71; however, the review by Xie and Wu did not take into account articles published after 2016 and also included the study by Ko et al., whose HR data reported in the meta-analysis refer to HNSCC [46]. For the other prognostic indices, the data reported are not yet fully definitive, and drawing clear conclusions is not yet possible due to the small number of studies included, which, in fact, represents the limit of this systematic review. However, the data from the meta-analysis are in line with both those published by Xie and Wu [34] and those from systemic reviews conducted on the prognostic value of miR-21 for HNSCC [24].

Furthermore, as reported in the prognosis research strategies (PROGRESS) [47], studies reviewed in this meta-analysis belong to type 2 (prognostic factor research); such kinds of studies aim to assess the specific association between specific factors and clinical outcomes. However, type 2 studies have some limits, for instance, they can just split cohorts of people on “high” and “low” risk but cannot weigh for the individual probability of patients to develop the outcome. For such a reason, a meta-analysis of type 2 studies is prodromic to the development of type 3 studies, so called, “prognostic model research”. At present, there are still no studies on miR-21 in association with OSCC in phase 3.

However, there are already phase 3 studies on miR-21 with interesting results, in fact, a study conducted by Voortman et al. determined whether the expression levels of a microRNA panel (miR-21, miR-29b, miR-34a/b/c, miR-155, and let-7a) can be used as prognostic or predictive biomarkers in patients who have participated in the International Adjuvant Lung Cancer Trial (IALT), one of the largest randomized trials conducted on adjuvant chemotherapy in patients with non-small cell lung cancer (NSCLC).

The results indicated a significant association only for miR-21 and OS with an unfavorable prognostic effect as a function of reduced expression. Furthermore, no other single or combinatorial microRNA expression profile predicted prognostic survival with adjuvant chemotherapy (cisplatin).

This may be an example of how the knowledge, in a diagnostic phase, of an altered tissue expression of miR-21 can give indications on the treatment as a function of an unfavorable prognosis [48].

In addition, many other miRs have landed in phase 3 studies, in fact, the results obtained from a multicenter retrospective study in which a series of serum miRNAs (miR-193a-3p, miR-369-5p, miR-672, miR-429, and let-7i) was identified in preclinical patients (average 6 months prior to clinical diagnosis) with hepatocellular carcinoma (HCC) with a high risk of developing 6–12 months later, after measurement [49]. This is an example of how the presence of the altered expression of serum miRs, combined with conventional screening tools, can predict and prevent HCC in a population at risk.

In another phase 3 study performed on patients with chronic lymphocytic leukemia (CLL), the disruption of the tumor suppressor pathway TP53 by downregulation of the 34° microRNA with deletion of 17p13 (del17p) to complete loss of TP53 function was associated with a poor prognosis [50].

The knowledge of the prognostic potential of miR-21 could help define the prognosis at the time of diagnosis, translating into clinical practice, in the presence of an OSCC that has an unfavorable prognostic signature at the histological diagnosis (with low OS, RFS), and could convey the treatment towards a more or less aggressive therapeutic and surgical approach, thus modulating the treatment according to the prognosis through the creation of individualized and tailor-made prognostic risk prediction models for the patient.

Furthermore, in the presence of a low OS, as well as a high incidence of relapses (low RFS) prognostically detected by the biomarker, there could be the basis for being able to approach new types of interventions with the aim of modifying the unfavorable course of the OSCC.

The prognostic capabilities of a biomarker such as miR-21 should, therefore, be investigated in many studies; meta-analyses in this perspective aim not only to aggregate the results but also to quantify and identify reporting biases and analysis deficiencies in primary studies. Leading to a greater understanding of how the prognostic biomarker can be used to improve clinical outcomes, clinical management of patients and the development of new treatments that lead to a more favorable course of the OSCC.

## 5. Conclusions

The data in the literature and the results emerging from the systematic review indicate that miR-21 can provide prognostic indications on oral carcinoma.

## Figures and Tables

**Figure 1 ijerph-19-03396-f001:**
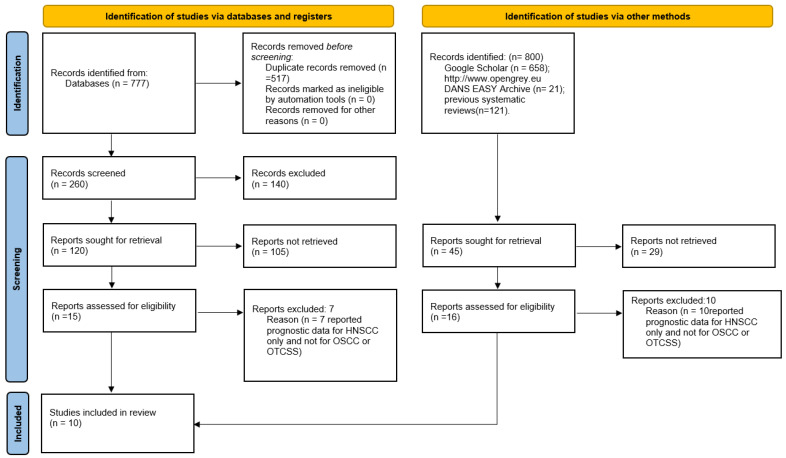
Flowchart of the different phases of the systematic review.

**Figure 2 ijerph-19-03396-f002:**
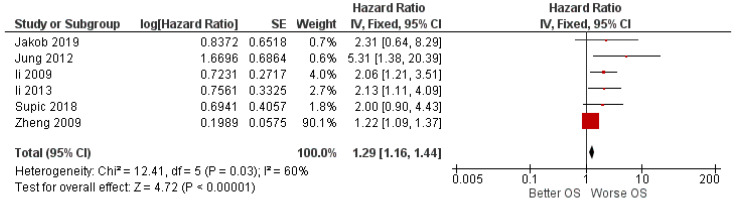
Forest plot of the fixed-effects model of the meta-analysis of the first outcome.

**Figure 3 ijerph-19-03396-f003:**
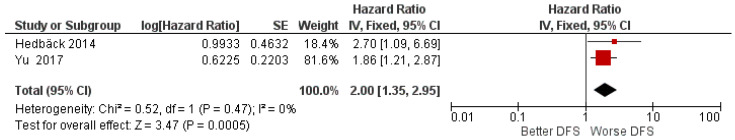
Forest plot of the fixed-effects model of the meta-analysis of the second outcome.

**Figure 4 ijerph-19-03396-f004:**
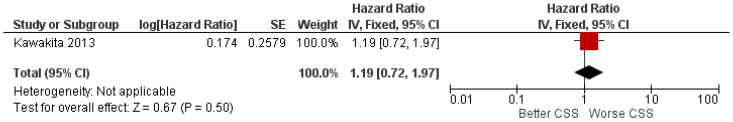
Forest plot of the fixed-effects model of the meta-analysis of the third outcome.

**Figure 5 ijerph-19-03396-f005:**
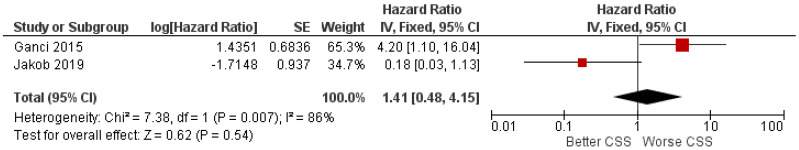
Forest plot of the fixed-effects model of the meta-analysis of the fourth outcome.

**Figure 6 ijerph-19-03396-f006:**
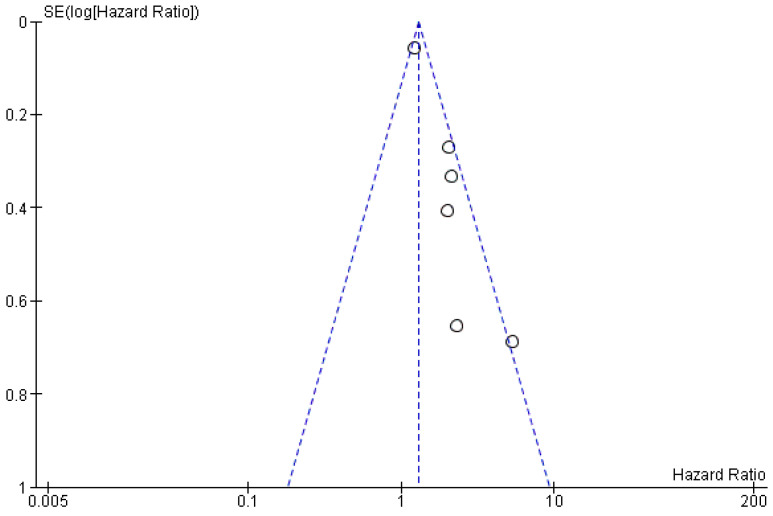
Funnel plot for the primary outcome; *I*^2^ = 60%.

**Table 1 ijerph-19-03396-t001:** Main data extracted from the studies.

First Author, Date	Country	Study Design	Number of Patients	Tumor Type/Tumor Site	Cut-Off	miR	HR miR-21 Low and High Expression (OS, CSS, DFS, RFS)
Jung (2012) [37]	USA	RT	17	OSCC	Median	miR-7, miR-21, miR-424	OS: HR, 5.31 (C.I.: 1.39–20.38)
Kawakita (2014) [38]	Japan	RT	79	OTSCC	Score ≥ 1	miR-21	CSS: HR, 1.19 (0.71–1.9)
Hedbäck (2014) [39]	Denmark	RT	86	OSCC (tongue or floor of the mouth)	Tertile	miR-21	DFS: HR, 2.70 (1.1–6.9)
Yu (2017) [40]	Taiwan	RT	100	OSCC (buccal mucosa tongue and mouth floor)	Tertile	miR-21	DFS: HR, 1.87 (1.21–2.87)
Supic (2018) [32]	Serbia	RT	60	Tongue carcinoma	ROC analysis, cut-off 9.38	miR-183, miR-21	OS: HR, 2.002 (0.904–4.434)
Jakob (2019) [33]	Germany	RT	36	OSCC	Median	miR-21, miR-29, miR-31, miR-99a, miR-99b, miR-100, miR-143, miR-155.	OS: HR, 2.31 (0.62–8.58) RFS: HR, 0.18 (0.02–139)
Li (2013) [41]	China	RT	63	OSCC	Score > 3	miR-21	OS: HR, 2.13 (1.11–4.10)
Zheng (2016) [42]	China	RT	72	Tongue	Score ≥ 2	miR-21	OS: HR, 1.22 (1.09–1.36)
Li (2009) [43]	China	RT	103	Tongue	Median	miR-21	OS: HR, 2.06 (1.21–3.51)
Ganci (2016) [44]	Italy	RT	92	OSCC	Signal score	miR-130b, miR-141, miR-21, miR-96	RFS: HR, 4.2(1.11–5.98)

**Table 2 ijerph-19-03396-t002:** Assessment of risk of bias within the studies.

First Author, Date	Sample	Clinical Data	Marker Quantification	Prognostication	Statistics	Classical Prognostic Factors	
Jung (2012) [37]	1	2	3	2	2	3	13
Kawakita (2014) [38]	3	2	2	2	1	3	13
Hedbäck (2014) [39]	3	3	2	3	3	2	16
Yu (2017) [40]	3	3	3	3	3	2	17
Supic (2018) [32]	2	3	3	3	3	3	17
Jakob (2019) [33]	1	3	3	3	3	3	16
Li (2013) [41]	2	2	3	2	2	3	14
Zheng (2016) [42]	3	2	3	3	2	2	15
Li (2009) [43]	3	3	3	3	3	3	18
Ganci (2016) [44]	3	2	3	3	3	2	16

**Table 3 ijerph-19-03396-t003:** Evaluation of GRADE pro-GDT.

Certainty Assessment							№ of Patients	Effect	Certainty
№ of Studies	Study Design	Risk of Bias	Inconsistency	Indirectness	Imprecision	Other Considerations		Relative (95% CI)	
6	Observational studies	Not serious	Not serious ^1^	Not serious	Serious	All plausible residual confounding would suggest spurious effect, while no effect was observed Dose–response gradient	351	HR: 1.29 (1.16 to 1.44)	⨁⨁⨁ ^2^ Moderate

^1^ In three studies, the HR value was extrapolated from the Kaplan–Meier curves reported in the manuscripts; this is not free from errors and can be a source of inaccuracy.^2^ Certainty: ⨁⨁⨁ moderate.

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
