# Peer review of "MicroRNA-21 Expression as a Prognostic Biomarker in Oral Cancer: Systematic Review and Meta-Analysis"

_ijerph, 2022, doi:10.3390/ijerph19063396_

Round 1
Reviewer 1 Report
Dear Authors,
thank you for submitting this review and meta-analysis paper. There certainly is a lack of studies in the field of biomarkers for early oral cancer diagnosis or prognostic evaluation. OC is painless and mostly arises at sites not causing symptoms or difficulties found by patient alone. Without systematic examinations at regular dental check-ups there is a risk for delayed discovery of advanced cancer leading to excessive surgical, radiological or chemotherapeutical treatment causing great postoperative morbidity with long rehabilitation period. The therapeutic decision is in many cases not straightforward. Research in prognostic biomarker options for more systematic treatment decision is clinically relevant. This review and meta-analysis was well done despite the lack of research available.
- Title: the title is clear and sufficiently summarizes the research topic.
- Abstract: it is short and clear.
- Introduction: sound, short and clear with sufficient information about the topic and previous research
- Methods: well described and they follow PRISMA protocols recommended for objective systematic review.
- Results: are explained well outlined and provides appropriate table and figure representation.
- Discussion: short, explanatory but need an addition of short explanation of results regarding biomarker potential clinical use and impact on treatment protocols.
- References: check the references in text. There are some inconsistencies with reference styles used in text.
Author Response
Dear Authors,
thank you for submitting this review and meta-analysis paper. There certainly is a lack of studies in the field of biomarkers for early oral cancer diagnosis or prognostic evaluation. OC is painless and mostly arises at sites not causing symptoms or difficulties found by patient alone. Without systematic examinations at regular dental check-ups there is a risk for delayed discovery of advanced cancer leading to excessive surgical, radiological or chemotherapeutical treatment causing great postoperative morbidity with long rehabilitation period. The therapeutic decision is in many cases not straightforward. Research in prognostic biomarker options for more systematic treatment decision is clinically relevant. This review and meta-analysis was well done despite the lack of research available.
- Title: the title is clear and sufficiently summarizes the research topic.
- Abstract: it is short and clear.
- Introduction: sound, short and clear with sufficient information about the topic and previous research
- Methods: well described and they follow PRISMA protocols recommended for objective systematic review.
- Results: are explained well outlined and provides appropriate table and figure representation.
- Discussion: short, explanatory but need an addition of short explanation of results regarding biomarker potential clinical use and impact on treatment protocols.
- References: check the references in text. There are some inconsistencies with reference styles used in text.
Answer
Dear reviewer,
thank you for your valuable suggestion, such comments revealed skilled knowledge in the field.
We have added a section in the manuscript explaining the clinical value of results. As reported in the prognosis research strategies (PROGRESS) (10.1371/journal.pmed.1001380), studies reviewed in this meta-analysis belong to type 2 (prognostic factor research), such kind of studies aim to assess the specific association between specific factors and clinical outcomes. However, type 2 studies have some limits, for instance, they can just split cohorts of people on "high" and "low" risk but cannot weight for the individual probability of patients to develop the outcome. For such a reason, meta-analysis of type 2 studies is prodromic to the development of type 3 studies, so called "prognostic model research" where biomarkers are coupled to other clinical factors to weight for the individual patient risk.
References: All references have been checked and correct.
Best Regards Mario Dioguardi
Reviewer 2 Report
There are typographical errors in the manuscript, which should be addressed prior to acceptance and publication. These include:
Page 2, Line 66 methyla-tion; no hyphen needed
Page 2, Line 67 (microRNA)); would suggest rewording as "...non-coding alterations of RNA including microRNAs (miRNA)."
Page 2, Line 94 - 98; Authors switch between IJERPH style of references with brackets, such as [XX} and ...recent studies by Supic et al. (2018)...and Jakob et al. (2019) with a mixture use in Line 98 ....by Jakob et al. [33]. Please use the IJERPH format with brackets throughout.
The Methods are very well described and follow PRISMA protocols closely. Data collection, extraction and verification by multiple users was reported. Good design.
The Results section is very well outlined and provides easy to read Tables, with appropriate data extraction and bias analysis. Forest plots with Hazard Ratio are excellent and authors should be commended for thorough analysis. Not sure that GRADE pro-GDT was absolutely necessary but complements the robust nature of this study.
Discussion and Conclusions might need a bit of reworking to remove the reference problems described above (e.g. Page 9, Line 308 Xie and Wu (2017 [34])...please use the IJERPH method of referencing throughout.
This study involved a systematic review and meta analysis regarding miR-21 as a prognostic indicator for oral cancer. It has interesting relevant, although prior research has demonstrated miR-21 expression in oral cancer, and systematic reviews have been conducted on microRNA expression and oral cancer prognosis, these were conducted in 2015 and 2018 and much has been added to the evidence base since that time regarding miR-21.
The evaluation of oral cancer expression of miR-21 is not original, but the evaluation of miR-21 as a stand alone prognostic indicator in a systematic review and meta analysis has not yet been published.
This study adds considerable context and data to the previous systematic reviews by Troiano et al., 2018 and Jamali et al., 2015. Both studies had found miR-21 expression (among others) was associated with poor prognosis, but the current study adds many more studies and patient outcomes to these previous observations - improving the inferential potential of this analysis.
The paper is well written with minor corrections noted previously. And the text is clear and easy to read with minor corrections noted previously.
The conclusions these are clearly supported by the evidence provided.
The authors clearly address the main question and provide a review consistent with the supporting evidence that adds to the understanding of miR-21 in oral cancer prognosis and epidemiology.
Author Response
There are typographical errors in the manuscript, which should be addressed prior to acceptance and publication. These include:
Page 2, Line 66 methyla-tion; no hyphen needed
Page 2, Line 67 (microRNA)); would suggest rewording as "...non-coding alterations of RNA including microRNAs (miRNA)."
Page 2, Line 94 - 98; Authors switch between IJERPH style of references with brackets, such as [XX} and ...recent studies by Supic et al. (2018)...and Jakob et al. (2019) with a mixture use in Line 98 ....by Jakob et al. [33]. Please use the IJERPH format with brackets throughout.
The Methods are very well described and follow PRISMA protocols closely. Data collection, extraction and verification by multiple users was reported. Good design.
The Results section is very well outlined and provides easy to read Tables, with appropriate data extraction and bias analysis. Forest plots with Hazard Ratio are excellent and authors should be commended for thorough analysis. Not sure that GRADE pro-GDT was absolutely necessary but complements the robust nature of this study.
Discussion and Conclusions might need a bit of reworking to remove the reference problems described above (e.g. Page 9, Line 308 Xie and Wu (2017 [34])...please use the IJERPH method of referencing throughout.
This study involved a systematic review and meta analysis regarding miR-21 as a prognostic indicator for oral cancer. It has interesting relevant, although prior research has demonstrated miR-21 expression in oral cancer, and systematic reviews have been conducted on microRNA expression and oral cancer prognosis, these were conducted in 2015 and 2018 and much has been added to the evidence base since that time regarding miR-21.
The evaluation of oral cancer expression of miR-21 is not original, but the evaluation of miR-21 as a stand alone prognostic indicator in a systematic review and meta analysis has not yet been published.
This study adds considerable context and data to the previous systematic reviews by Troiano et al., 2018 and Jamali et al., 2015. Both studies had found miR-21 expression (among others) was associated with poor prognosis, but the current study adds many more studies and patient outcomes to these previous observations - improving the inferential potential of this analysis.
The paper is well written with minor corrections noted previously. And the text is clear and easy to read with minor corrections noted previously.
The conclusions these are clearly supported by the evidence provided.
The authors clearly address the main question and provide a review consistent with the supporting evidence that adds to the understanding of miR-21 in oral cancer prognosis and epidemiology.
Answer
Dear Reviewer,
thanks for your valuable suggestions, such comments revealed a qualified knowledge in the field.
All suggested corrections have been made to the manuscript.
All references have been modified according to the reference style of the journal.
Best Regards Mario Dioguardi.
Reviewer 3 Report
The meta-analysis and the entire study were performed very prudently and completely. There is no doubt about the scientific integrity of the study.
What I find very lacking, however, is the clinical interpretation of the results. What advice can you give to an oral surgeon practicing oncology? Should the miRNA profile of every patient be done? At what cut-off value is there now an increased risk for the patient? And what does increased risk mean? Does it mean death, or occurrence of a recurrence, or metastasis? Is it possible to give a specific therapy recommendation based on the miRNA values?
In principle, a complete section is missing here, both in the results and in the discussion.
Author Response
The meta-analysis and the entire study were performed very prudently and completely. There is no doubt about the scientific integrity of the study.
What I find very lacking, however, is the clinical interpretation of the results. What advice can you give to an oral surgeon practicing oncology? Should the miRNA profile of every patient be done? At what cut-off value is there now an increased risk for the patient? And what does increased risk mean? Does it mean death, or occurrence of a recurrence, or metastasis? Is it possible to give a specific therapy recommendation based on the miRNA values?
In principle, a complete section is missing here, both in the results and in the discussion.
Answer
Dear reviewer,
thank you for your valuable suggestion, such comments revealed skilled knowledge in the field.
We have added a section in the manuscript explaining the clinical value of results. As reported in the prognosis research strategies (PROGRESS) (10.1371/journal.pmed.1001380), studies reviewed in this meta-analysis belong to type 2 (prognostic factor research), such kind of studies aim to assess the specific association between specific factors and clinical outcomes. However, type 2 studies have some limits, for instance, they can just split cohorts of people on "high" and "low" risk but cannot weight for the individual probability of patients to develop the outcome. For such a reason, meta-analysis of type 2 studies are prodromic to the development of type 3 studies, so called "prognostic model research" where biomarkers are coupled to other clinical factors to weight for the individual patient risk.
Best Regards Mario Dioguardi
Reviewer 4 Report
I consider that the manuscript presented by Dioguardi and collaborators shows interesting data of potential interest for scientific community.
Overall the work is well-explained and well-presented.
The number of studies is few but the analysis is robust and considers a big number of patients.
I would accept the manuscript in the present form.
Author Response
I consider that the manuscript presented by Dioguardi and collaborators shows interesting data of potential interest for scientific community.
Overall the work is well-explained and well-presented.
The number of studies is few but the analysis is robust and considers a big number of patients.
I would accept the manuscript in the present form
Answer
Thank you for reviewing the manuscript and for the comments.
Best regards Mario Dioguardi
Round 2
Reviewer 3 Report
I find the authors' answer insufficient. Even if only phase 2 studies are summarised here, the present meta-analysis must nevertheless provide an outlook. Are there already phase 3 studies on miRNAs? What diagnostic, prognostic or therapeutic consequences will there be? When an oral surgeon reads this article, what added value in terms of information will he or she take away? Unfortunately, 50% of patients with oral squamous cell carcinoma still die. Can miRNAs perhaps bring an improvement in prognosis here?
For me, this is a key question that needs to be answered, because otherwise what good is a meta-analysis anyway?
Author Response
Reviewer
I find the authors' answer insufficient. Even if only phase 2 studies are summarised here, the present meta-analysis must nevertheless provide an outlook. Are there already phase 3 studies on miRNAs? What diagnostic, prognostic or therapeutic consequences will there be? When an oral surgeon reads this article, what added value in terms of information will he or she take away? Unfortunately, 50% of patients with oral squamous cell carcinoma still die. Can miRNAs perhaps bring an improvement in prognosis here?
For me, this is a key question that needs to be answered, because otherwise what good is a meta-analysis anyway?
ANSWER
Dear reviewer,
His comments and suggestions were very helpful in improving understanding of the manuscript and the role miR-21 could play in the clinical and surgical practice of treating OSCC.
At present there are still no studies on miR-21 in association with OSCC in phase 3;
However, there are already phase 3 studies on miR-21 with interesting results, in fact a study conducted by Voortman et al. determined whether the expression levels of a microRNA panel (miR-21, miR-29b, miR-34a / b / c, miR-155 and let-7a), can be used as prognostic or predictive biomarkers in patients who have participated in the International Adjuvant Lung Cancer Trial (IALT), one of the largest randomized trials conducted on adjuvant chemotherapy in patients with non-small cell lung cancer (NSCLC).
The results indicated a significant association only for miR-21 and OS with an unfavorable prognostic effect as a function of reduced expression. Furthermore, no other single or combinatorial microRNA expression profile predicted prognostically survival with adjuvant chemotherapy (cisplatin).
This may be an example of how the knowledge, in a diagnostic phase, of an altered tissue expression of miR-21 can give indications on the treatment as a function of an unfavorable prognosis [1].
In addition, many other miRs have landed in phase 3 studies, in fact the results obtained from a multicenter retrospective study in which a series of serum miRNAs (miR-193a-3p, miR-369-5p, miR-672, miR-429 and let-7i) was identified in preclinical patients (average 6 months prior to clinical diagnosis) with hepatocellular carcinoma (HCC) with a high risk of developing 6-12 months later after measurement [2]. Example of how the presence of altered expression of serum miRs, combined with conventional screening tools, can predict and prevent HCC in a population at risk.
In another phase 3 study performed on patients with chronic lymphocytic leukemia (CLL), the disruption of the tumor suppressor pathway TP53 by downregulation of the 34 ° microRNA with deletion of 17p13 (del17p) to complete loss of TP53 function was associated with a poor prognosis [3].
The knowledge of the prognostic potential of miR-21 could help define the prognosis at the time of diagnosis, translating into clinical practice, in the presence of an OSCC that has an unfavorable prognostic signature at the histological diagnosis (with low OS, RFS), could convey the treatment towards a more or less aggressive therapeutic and surgical approach, thus modulating the treatment according to the prognosis through the creation of individualized and tailor-made prognostic risk prediction models for the patient.
Furthermore, in the presence of a low OS as well as a high incidence of relapses (low RFS), prognostically detected by the biomarker, there could be the basis for being able to approach new types of interventions with the aim of modifying the unfavorable course of the OSCC.
The prognostic capabilities of a biomarker such as miR-21 should therefore be investigated in many studies; meta-analyzes in this perspective aim not only to aggregate the results but also to quantify and identify reporting biases and analysis deficiencies in primary studies. Leading to a greater understanding of how the prognostic biomarker can be used to improve clinical outcomes, clinical management of patients and the development of new treatments that lead to a more favorable course of the OSCC.
Best regards Mario Dioguardi
- Voortman, J.; Goto, A.; Mendiboure, J.; Sohn, J.J.; Schetter, A.J.; Saito, M.; Dunant, A.; Pham, T.C.; Petrini, I.; Lee, A., et al. MicroRNA expression and clinical outcomes in patients treated with adjuvant chemotherapy after complete resection of non-small cell lung carcinoma. Cancer Res 2010, 70, 8288-8298, doi:10.1158/0008-5472.can-10-1348.
- Li, L.; Chen, J.; Chen, X.; Tang, J.; Guo, H.; Wang, X.; Qian, J.; Luo, G.; He, F.; Lu, X., et al. Serum miRNAs as predictive and preventive biomarker for pre-clinical hepatocellular carcinoma. Cancer letters 2016, 373, 234-240, doi:10.1016/j.canlet.2016.01.028.
- Dufour, A.; Palermo, G.; Zellmeier, E.; Mellert, G.; Duchateau-Nguyen, G.; Schneider, S.; Benthaus, T.; Kakadia, P.M.; Spiekermann, K.; Hiddemann, W., et al. Inactivation of TP53 correlates with disease progression and low miR-34a expression in previously treated chronic lymphocytic leukemia patients. Blood 2013, 121, 3650-3657, doi:10.1182/blood-2012-10-458695.